# The GOGIRA System: An Innovative Method for Landslides Digital Mapping

Michele Licata  and Giandomenico Fubelli *

Department of Earth Sciences, University of Turin, Via Valperga Caluso, 35, 10125 Turin, Italy
* Correspondence: giandomenico.fubelli@unito.it

**Abstract:** Landslide mapping techniques have had many improvements in recent decades, the main field of development has been on traditional cartographic techniques and to a lesser extent on indirect numerical cartography. As for Direct Numerical Cartography (DNC), only a few improvements have been made due to the complexity and economic cost of the new technologies. To meet this lack in DNC techniques GOGIRA (Ground Operative-system for GIS Input Remote-data Acquisition), a new system following the GIS (Geographic Information System) scheme, was developed. It is a suite of hardware and software tools, algorithms, and procedures for easier and cheaper DNC. Initial tests conducted on the Quincinetto landslide system (north-western Italy) demonstrated good results in terms of morphometric coherence and precision. A geomorphological map made with GOGIRA was compared with a highly detailed geomorphological map developed with modern tested methods. In conclusion GOGIRA proved to be a valid system for geomorphological DNC when applied to a complex landslide system, considering the early stage of developing results for linear and point mapping was excellent, as for polygonal elements more studies must be conducted to improve accuracy and precision.

**Keywords:** digital apping; geomorphology; Arduino; GIS; numerical cartography; landslide; geomorphological mapping; remote sensing; Italy

## 1. Introduction

The first geomorphological map was made in 1914 [1] however modern geomorphological mapping concepts were only initiated in the early 1950s [2]. Initially, mapping techniques were developed with particular attention to geomorphological legends which play a key role in representing the elements of crucial importance to geomorphologists. Information technology development played a fundamental role [3] laying the foundations of modern geomorphological Geographical Information Systems (GIS) [4] based on a multiscale digital spatial database approach [5,6].

Numerical Cartography (NC) [7] has been relatively slow to improvements because of its dual nature: the first consists of digitalization of "traditional" paper-made maps, called Indirect Numerical Cartography (INC); the second one Direct Numerical Cartography (DNC) [8], is done by collecting "digital-born" data directly during fieldwork.

The first approach requires an additional step in the workflow to digitalize field maps and to compile the attribute tables. This is a time-consuming activity that may introduce error propagation due to the digitalizing procedure of scanning, georeferencing and shape re-drawing. With the DNC approach, the fieldwork mapping activity is done directly in digital format, which can avoid the redundancy of the digitalization process and can reduce the propagation of cartographic errors, making the mapping process faster, and with known instrumental accuracy and precision [7].

Data in DNC can be collected through several sources and tools. An interesting classification proposed by Guzzetti et al. [9] distinguishes three macro-categories: (i) morphometric analysis through a Digital Terrain Model (DTM), (ii) satellite imagery analysis, and (iii) digital acquisition tools.

DTMs have become a standard tool in geomorphological NC [10], it consists of raster data where every pixel has a 2D spatial dimension (in meters) and the associated value represents the elevation (third dimension). Currently there are two main techniques to make a DTM: photogrammetry or LiDAR (Light Detection And Ranging) [11–13]. Photogrammetry utilizes an aerial vector such as an airplane, helicopter, or drone, and in some applications also a terrestrial (hand camera or terrestrial vehicle) mode. It is a low-cost technology but with a high computational cost. The second technique is LASER technology based; which can result in a high-resolution DTM. Also, in this second case raw data must be processed to obtain a reliable Digital Terrain Model that faithfully represents the real-world topography.

From DTM data it is possible to not only obtain elevation information but also other useful data such as hillshade, slope, aspect, or curvature. Many GIS tools use DTMs in their computational algorithms. In the last few years, developments in technologies such as optical cameras, LASERs, drone vectors and computers has facilitated the production of high-resolution DTMs accessible to geomorphologists [2].

Satellite imagery is raster data that contains radiometric measures of light spectrum, commonly in the visible (RGB), Thermal Infra-Red (TIR), and the microwave spectrums. RGB images are a useful tool in digital photo interpretation of geomorphologic elements due to the differences in texture (e.g., soil, vegetation, water, rock) [7,14–16]. Imagery cover is almost worldwide, often with multi-temporal series. By combining RGB and TIR images it is possible to perform a semi-automatic classification of the territory. This is possible due to spectral signatures which allow the distinction of different materials and physical conditions such as river turbidity, snow versus ice detection, health of vegetation, soil denudation, or soil water saturation [17]. Finally, the microwave spectrum has an interesting application in landslide monitoring. The PS-inSAR (Persistent Scatterer interferometry Synthetic Aperture Radar) [18] technique is used to measure millimetric ground movement. It is suitable for landslide analysis where the landscape demonstrates local changes in surface position elements [19].

The third macro-category according to Guzzetti et al. [9] is "digital acquisition tools". For the purpose of this paper, a general definition is proposed based on the purpose of the tools: "angle and distance measures", "pointcloud acquisition", and "visualization and attributes input". Commonly used cartographic tools belonging to the first category are total station, theodolite, or LASER rangefinder. Some authors tested rangefinder binoculars to produce a landslide inventory [8,20,21]. This application showed interesting results but due to the high price of the device, there is little research using this technology.

Point cloud acquisition techniques include technologies that collect georeferenced point data. The main example is the GNSS (Global Navigation Satellite System) that uses satellite technology to obtain the coordinates of the ground receivers. Depending on the type of receiver (single or double frequency), GNSS precision can vary from meters to centimeters [20], and is useful for any geomorphological mapping that has direct access and good satellite coverage. Other pointcloud technologies are "photogrammetry" and "laser scanning". As explained previously, these are useful to develop DTM models and 3D models of slopes, rock walls or other surface elements. These are very useful technologies due to the flexibility of possible tasks, however the cost in terms of data computation of large datasets (photogrammetry) and device price (laser scanner) is a limitation to commonly using geomorphologic tools [21,22].

"Visualization and attributes input" devices are intended for all electronic devices used in data consulting and digital inputs. Common devices are PCs, smartphones, tablets, and PDAs (Personal Digital Assistant). In the late 1990s, technology wasn't sufficiently developed for their practical use. In recent years, many authors have recognized their reliability for geological digital mapping [20,23,24], but there are concerns about the accuracy of the integrated sensors accuracy (GPS, accelerometer, magnetometer, gyroscope), battery autonomy, and the technical informatic skills of mappers [25].

This paper will introduce a new method for DNC, developed with the purpose of making direct digital mapping tools more accessible to users. With this purpose GOGIRA

(Ground Operative-system for GIS Input Remote-data Acquisition) was developed and tested in the Quincinetto landslide system (north-western Italy). GOGIRA, currently in a prototype phase, consists of a suit of hardware and software tools. Two devices for digital data acquisition were developed using low-cost and open-source technologies; a Central Unit (CU) for data storing and attribute input; a Python algorithm is used to process the collected data; and a semi-automatic legend for QGIS projects [26] were performed.

The CoordFinder algorithm uses a metrical system for data elaboration, therefor the WGS84 (World Geodetic System 1984) ellipsoid with the UTM (Universal Transverse Mercator) datum was chosen [27,28]. This allows for use of the GOGIRA system almost worldwide as long as a DTM is available and the area is covered by GNSS signal. DTM resolution, GNSS precision, and field survey conditions (see section) must be considered to obtain strong results.

## 2. Materials and Methods

The analysis of previous research has shown a gap in the DNC techniques. The main problems are: (i) the high skills required for a mapper to use devices and methods, (ii) the high cost of devices, and (iii) the difficulty of interfacing different devices/software. Furthermore, DNC is preferred by most high-skilled mappers because the devices and software available are very sophisticated and often geomorphologists must adapt them to the DNC geomorphological mapping survey.

GOGIRA system was designed to fill this gap, making an interconnected system utilizing open source and low-cost technologies. Devices and software where though to be easy-to-use and dedicated to filed mapping survey, with particular attention to geomorphological mapping.

### 2.1. GOGIRA as GIS System

The working principle of GOGIRA is based on a GIS system. A GIS is a complex system composed of procedures, software, hardware, data, and personnel (Figure 1); that allow one to insert, manipulate, analyze, and display data and georeferenced spatial information [4]. From this definition it is evident that a GIS system is more complex than mere software used for geospatial data management/visualization. The GOGIRA system was designed to follow the GIS structure to optimize the DNC process.

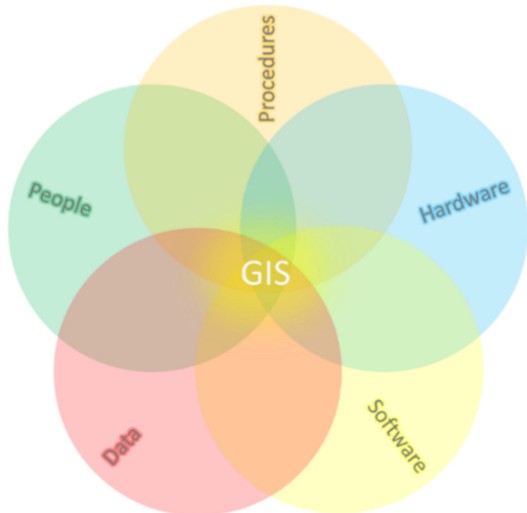

**Figure 1.** GIS system diagram (modified from [4]).

The system presented in this paper is a suite of hardware, software, and procedures to perform DNC with low-cost devices and software, and minimal time-consuming procedures. In the current state of development, it consists of:

- CoordFinder, an algorithm to convert local spherical coordinates into a cartographic reference system;
- Two devices, UGO (User-based Geomorphic Observer) and Range-R (Remote Rangefinder), designed for target local spherical coordinates acquisition;
- MARVIN, a smartphone Android application developed to collect UGO and Range-R data, set information about morphogenetic processes, and type of morphometry;
- A semi-automatic cartographic import and legend procedure for QGIS projects.

*2.2. Prototypes*

2.2.1. CoordFinder Algorithm

One of the main features of GOGIRA is the capability to find the cartographic coordinates of a physical target in the landscape. This is the role of the CoordFinder algorithm, developed in the Python programming language.

For the structure of the algorithm itself, all cartographic coordinates are expressed in the WGS84/UTM reference system, this is necessary because the software requires metric rather than angular coordinates. Considering that the field test area is in northwest Italy, the coordinate system chosen was the WGS84/UTM Zone 32N.

Inputs required are (i) the global cartographic coordinates of the mapper in WGS84/UTM Zone 32N, (ii) the local spherical coordinates of the target as azimuthal and zenithal angles (A), (iii) a DTM covering the area of the mapping survey, (iv) DTM resolution [m] and (v) extent in WGS84/UTM Zone 32N.

The operating principle of the algorithm is based on the intersection of the "Line Of Sight" (LOS) projected by the mapper's position coordinates through the sighted target (Figure 2A,B), with the DTM virtual land surface (Figure 2C).

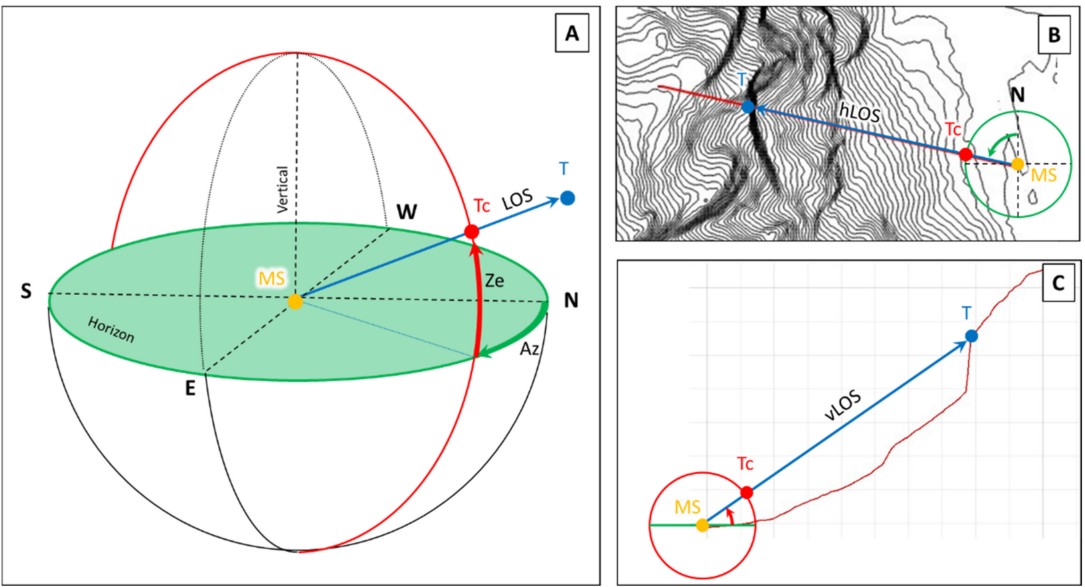

**Figure 2.** CoordFinder schematized working principle where (**A**) is the Pseudo 3D view, (**B**) the Map view, and (**C**) the Profile view. The green plane is the horizontal and the red circle is the vertical plane. (MS) Measuring Station, (T) Target sighted, (LOS) Line Of Sight, (Tc) Target projection on the vertical circle, (Ze) Zenithal angle, (Az) Azimuthal angle.

LOS is first calculated on the 2D horizontal plane (Equation (1)). The target point is projected onto a circle with (i) the East-North mapper coordinates as origin, (ii) arbitrary radius, and (iii) rotated clockwise from north by the azimuth angle (Figure 2A,B). The "m" $hLOS_N$ coordinates are calculated for the "n" $hLOS_E$ values, where "m" and "n" are respectively the north and east size of DTM. $tc_{(E,N)}$ are the east and north target point coordinates on the horizontal circle; $mp_{(E,N)}$ are the mapper east and north coordinates.

$$hLOS_{N(E)} = \frac{tc_N - mp_N}{tc_E - mp_E} \times (hLOS_E - mp_E) + mp_N \tag{1}$$

The vertical component of LOS (vLOS) is calculated similarly to hLOS. The target point is projected on a semicircle with: (i) origin set to x = 0, and y = "mapper elevation"; (ii) arbitrary radius; (iii) counterclockwise rotation from the horizontal by zenith angle (Figure 2C). The $vLOS_Z$ straight-line is calculated for the "x" $vLOS_X$ values (Equation (2)), where "x" corresponds to $hLOS_E$ number of values. $tc_{z,x}$ are the elevation and distance target point coordinates on the vertical semicircle; $mp_z$ is the mapper elevation coordinate.

$$vLOS_{Z(x)} = \frac{tc_z - mp_z}{tc_x} \times vLOS_x + mp_z \tag{2}$$

A topographic profile (Figure 2C) is made using DTM elevation data along $hLOS_{N(E)}$. The starting point is the mapper position, and the end point is the last pixel of the DTM along the profile line.

The difference between elevation values of vLOSz and topographic profile return the "delta" curve. The first minimum value, excluding the mapper position, is the intersection point between LOS and DTM, this point corresponds to the target sighted. From the target cell indexes and the DTM's georeferencing matrix, cartographic target coordinates are obtained in WGS84/UTM Zone 32N.

The CoordFinder algorithm is iterated for each target point of a mapped geomorphological shape, returning a list of georeferenced coordinates that are saved to a text file (.txt) in Well Known Text (WKT) format [29]. WKT is an Open Geospatial Consortium standard to represent spatial data in a textual format and the QGIS import procedure is quite simple and fast (see Section 2.3.4). Moreover, attributes such as morphogenetic agent and geomorphological shape code can be saved in the same text file in Comma Separated Values (CSV) format. It is possible to distinguish attributes from geometry in the import procedure, described in Section 2.3.4 for QGIS 3.16.

It is essential to highlight the importance of the cartographic reference system chosen for the algorithm. The UTM system is a metric reference system, which is fundamental for the calculus of the algorithm itself. To test CoordFinder in a different region of the globe where other reference systems are needed, one must use a metric reference system for all the georeferenced data of the project.

### 2.2.2. Tools

DNC need tools to transpose 3D real-world objects into a 2D digital map representation [7]. In the GOGIRA system this can be done using MARVIN CU with UGO and Range-R spherical coordinate acquisition devices.

UGO and Range-R are prototypes based primarily on Open-Source (OS) technology [30]. This allows for easy development and modification of the devices with low-cost sensors and components. Both devices were developed using a Arduino Nano board (Table 1), a robust OS microcontroller board for reliable electronic project development [31–36]. It mounts the ATmega328 microcontroller and has a series of analog and digital input/output pins, as well as a useful I2C (Inter Integrated Circuit) and SPI (Serial Peripheral Interface) communication protocol, which allows interfacing various sensors. For both devices, the software was developed with the C++ programming language. Third-party software libraries were used to interface the OS Arduino Software IDE (Integrated Development Environment) and the Arduino microcontroller (Table 2).

**Table 1.** Component list.

| Code | Description | Main Features | Price [1] [€/cad] |
|---|---|---|---|
| P160 | Rotary potentiometer | Resistance: 10 [KΩ]; Total mechanical travel: 300 [°] ± 20%; Temperature range: −20 to +70 [°C]. | ~0.5 |
| KY-040 | Rotary encoder | Pulses on 360° rotation: 20; 2-bit Gray code; Push button; Temperature range: −10 to +65 [°C]. | ~2 |
| HC-05 | Bluetooth SPP module | UART interface; baud rate from 9600 to 460,800; Integrated antenna; 3 [Mbps] Modulation 2.4G [Hz]; −80 [dBm] sensitivity; Temperature range: −20 to +70 [°C]. | ~7.5 |
| ATmega328 | Arduino nano board | Flash memory: 32 [KB]; SRAM: 2 [KB]; Clock speed: 16 [MHz]; DC current (I/O): 40 [mA]; Digital I/O pins: 22; Analog I/O pins: 8; Communication: I2C, SPI; Temperature range: −25 to +70 [°C]. | ~8 |
| MPU6050 | 6-axis Motion Tracking | 16-bit resolution triaxial accelerometer and gyroscope; operating currents: 3.6 [mA] (gyroscope), 0.5 [mA] (accelerometer); I2C communication; temperature range: −25 to +70 [°C]. GYROSCOPE: angular rete ±250 to ±2000 [°/s]; data output rate 8 [KHz]. ACCELEROMETER: full-scale range ±2 to ±16 [g]; data output rate 1 [KHz] | ~6 |
| BNO055 | 9-axis Absolute orientation | Triaxial gyroscope, accelerometer and magnetometer; temperature sensor; sensor-fusion modes; autocalibration mode; I2C communication; temperature range: −25 to +70 [°C]. GYROSCOPE: angular rete ±125 to ±2000 [°/s]; data output rate 100 [Hz]. ACCELEROMETER: full-scale range ±2 to ±16 [g]; data output rate 100 [KHz]. MAGNETOMETER: full-scale range: ±1200 [μT] (x,y), ±2000 [μT] (z); data output rate 20 [Hz]. | ~20 |
| SSD1306 | CMOS OLED display | 128x64 pixel resolution; I2C communication; 2 to 24 [mA] consumption. | ~5 |
| — | SD card module | SPI Communication; FAT16 or FAT32 formatting; SD card supported: 2 [GB]. | ~4 |

[1] Prices were evaluated on Amazon®, in date 19 July 2022.

**Table 2.** Library used.

| Name | Language | Reference |
|---|---|---|
| SPI | C++ | [37] |
| Wire | C++ | [38] |
| Adafruit_GFX | C++ | [39] |
| Adafruit_SSD1306 | C++ | [40] |
| SoftwareSerial | C++ | [41] |
| Kalman | C++ | [42] |
| SD | C++ | [43] |
| numpy | Python 3.9 | [44] |
| matplotlib | Python 3.9 | [45] |
| math | Python 3.9 | [46] |
| tkinter | Python 3.9 | [47] |
| csv | Python 3.9 | [48] |
| PIL | Python 3.9 | [49] |
| os | Python 3.9 | [50] |

UGO was designed to be cheap and reliable; it is a tripod-based device that can obtain measures of azimuth and zenith angles of a real-world sighted target (Figure 2). Components are chosen to be inexpensive and reliable to create a simple yet robust device to solve a single task with minimum waste of resources. The components list is:

- n° 2 Rotary potentiometer;
- n° 1 Rotary encoder;
- n° 1 HC-05 Bluetooth module;
- n° 2 Arduino Nano microcontroller;
- n° 1 MPU6050 Inertial Motion Unit (IMU);
- n° 1 SSD1306 OLED display;
- n° 1 SD-card module;
- n° 1 Toroidal bubble-level;
- n° 1 9-volt battery cell;
- n° 1 Red-dot optical sight;
- n° 1 Tripod.

A fixed, static body is mounted on the tripod. The main body can twist around a rotational axis orthogonal to the horizontal plane. All electronical components, a toroidal bubble-level, and a 9-volt external battery cell are mounted on the main body. The optical sight twists around a second rotational axis perpendicular to the first one and jointed to the main body (Figure 3).

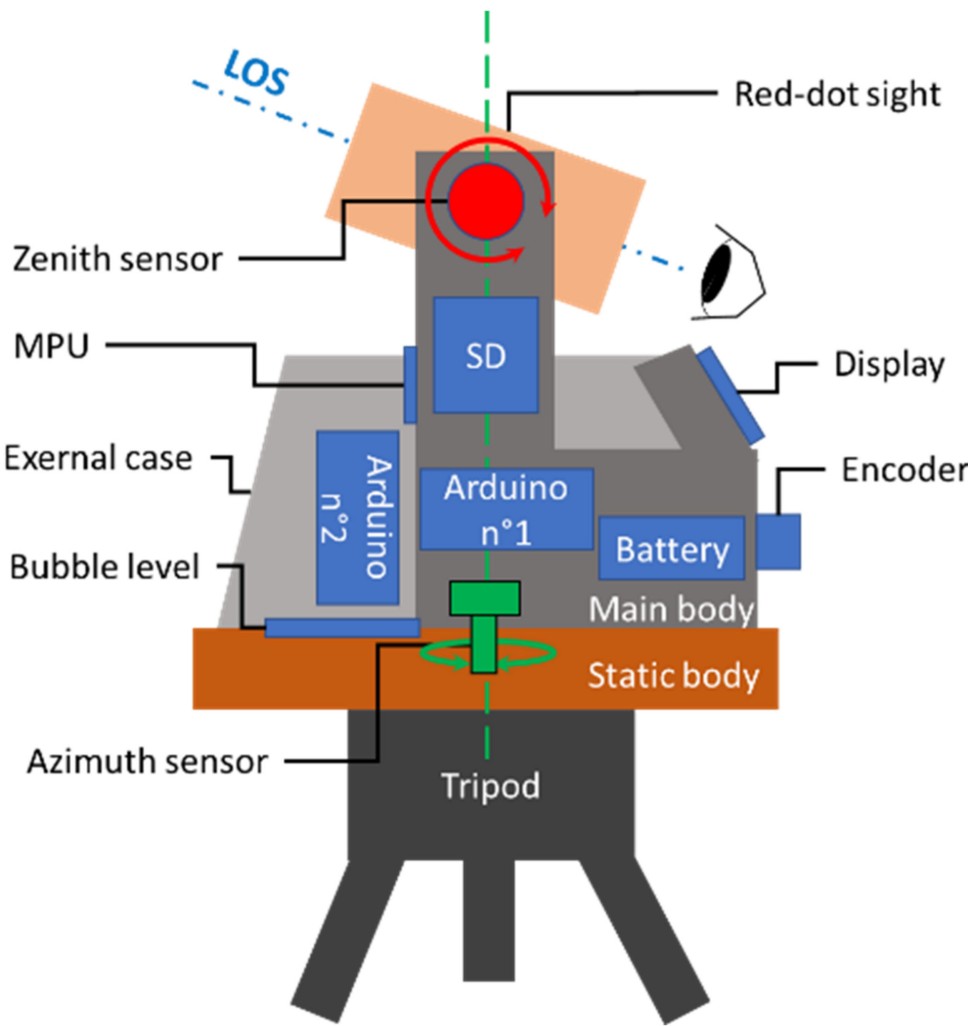

**Figure 3.** UGO components schematics.

The working principle is simple: the optical sight twists around the horizontal axis ($\pm 50°$ from horizontal) and the vertical axis ($300° \pm 10°$). Angle values (in degrees) are calculated by the product of voltage values read on the rotary potentiometer's pins, and the conversion factor specific for each resistor (Figure 4).

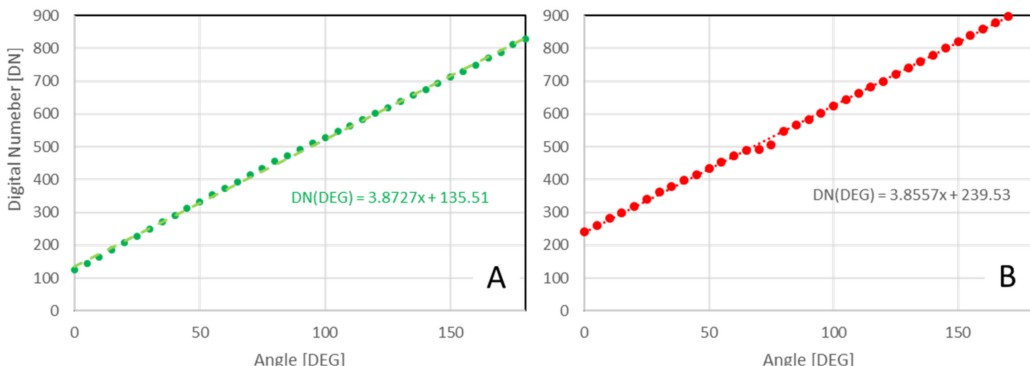

**Figure 4.** Potentiometer calibration of (**A**) azimuth potentiometer and (**B**) zenith potentiometer, made for angles between 0 and 180 [DEG] with 5 [DEG] step. The digital number (DN) is the voltage value read on the potentiometer's pins and converted into a digital value by the Arduino Nano internal 12-bit resolution ADC (Analog to Digital Converter).

Two Arduino Nano boards were wired in serial SPI communication to extend the total memory available; this was necessary because of their small RAM (Table 1), and the total memory required by the software and modules. The "master" Arduino board is responsible for the angle data read, the user input commands and the control display. The "slave" Arduino board receives the data from the "master", saves it on an SD card as a new record in ".txt" format, and sends it to MARVIN CU using the HC-05 Bluetooth communication module.

For UGO's calibration procedure (see Section 2.3.1), a double system was performed: the analog toroidal bubble-level and the digital system based on MPU6050 6-axes IMU. The MPU6050 calibration was made for the specific chip mounted on the prototype (Figure 5A,B). Pitch and roll angles are evaluated using a Kalman filter that combines accelerometric and gyroscopic values in a fusion-data algorithm [51].

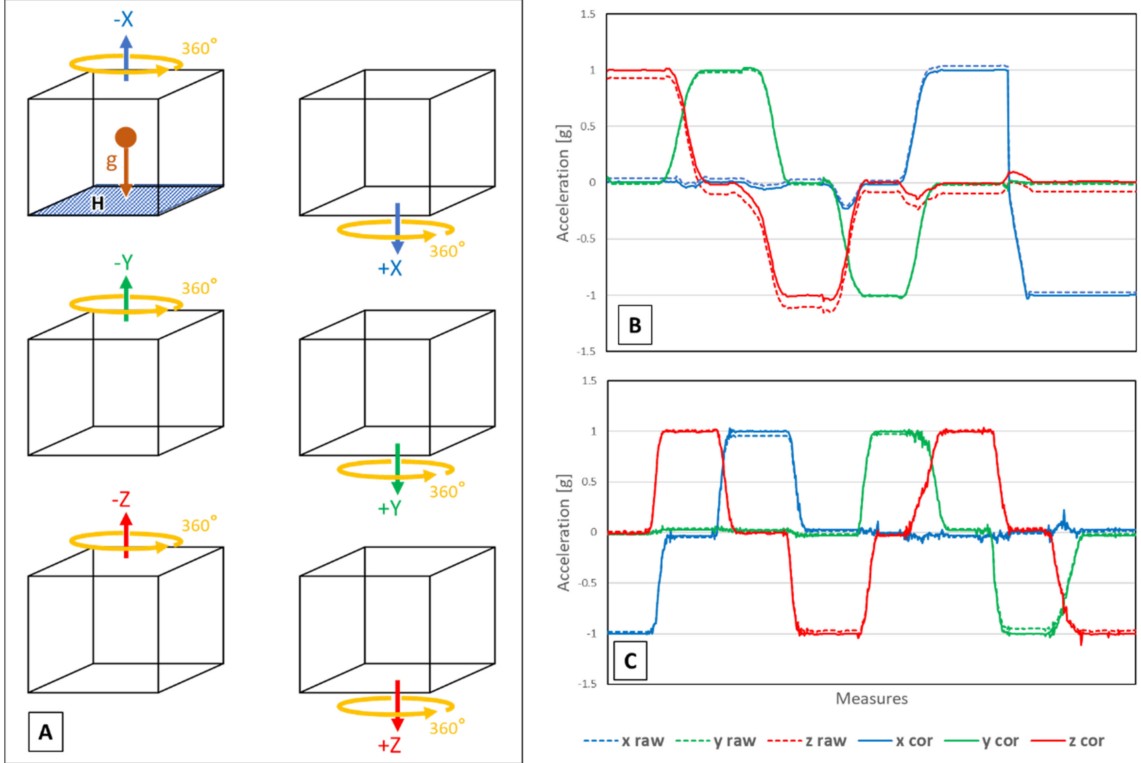

**Figure 5.** Accelerometer calibration. (**A**) Calibration procedure. (**B**) MPU6050. (**C**) BNO055.

Range-R is a device thought to solve a similar task to UGO but with significantly different features. Instead of azimuth and zenith, Range-R obtains heading angle (direction from magnetic north) and pitch angle (inclination of sight axis with respect to the horizontal plane) of the sighted target. With the BNO055 sensor data, a Nine Degree Of Freedom (NDOF) system [52] was made. This allows for free movement of Range-R in the 3D space, and no tripod is required as is the case for UGO. The component list is quite simple and consists of:

- n° 1 Arduino Nano microcontroller;
- n° 1 BNO055 Attitude and Heading Reference System (AHRS)
- n° 1 SSD1306 OLED display;
- n° 1 HC-05 Bluetooth module;
- n° 1 optical sight 10x zoom;
- n° 1 9-volt battery cell.

The BNO055 allow to provide an AHRS (Attitude and Heading Reference System): the triaxial magnetometer, gyroscope and accelerometer values are processed by the fusion-data built-in algorithm [53] to obtain pitch, roll and heading (Figure 6) values. The fusion-data algorithm is "Bosch Sensortec", a proprietary software wherein source-code is unavailable, so it was used as "black box" providing output heading, pitch, roll and calibration status.

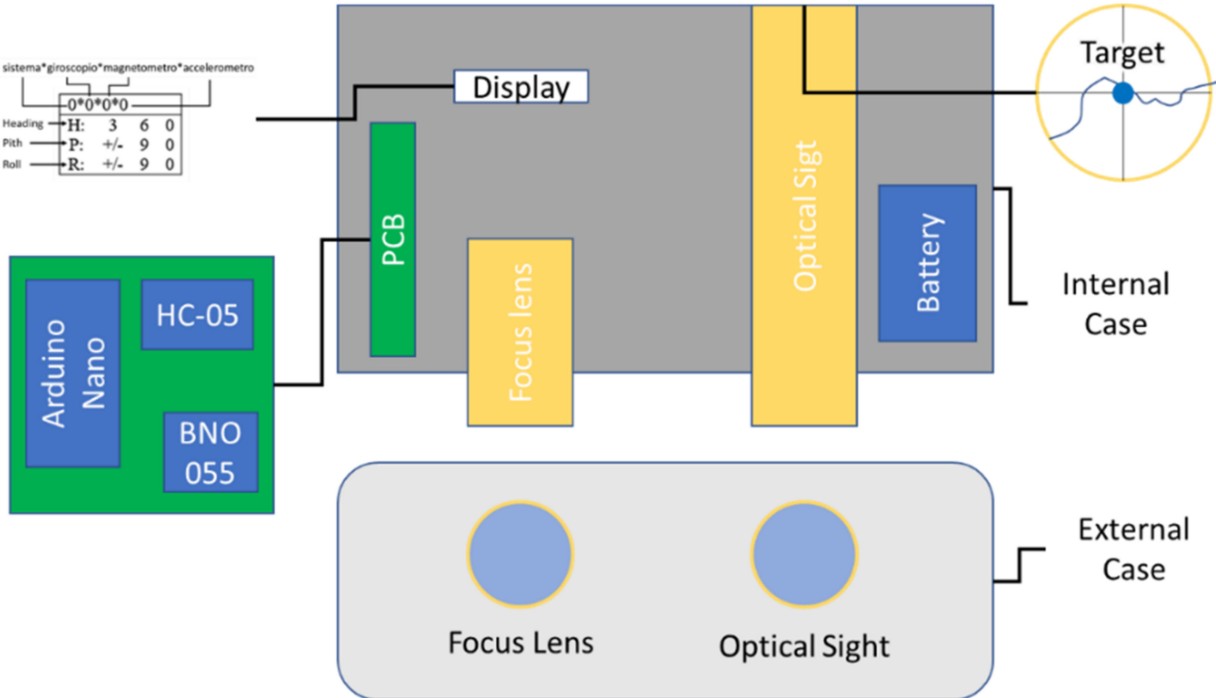

**Figure 6.** RANGER components schematics.

Sensor calibration (see Section 2.3.1) requires only a few seconds. Gyroscope and magnetometer calibration is easy, but the flat horizontal surface required for accelerometer calibration (Figure 5A) can be difficult to find in natural environments. To overcome this problem, the calibration procedure was performed during the device assembly phase and offset values (Figure 5C) are set automatically without further procedures by the user. Calibration status is given by BNO055 internal software as a numeric variable from "0" (system is non-calibrated) to "3" (fully calibrated) for each triaxial sensor, and for the whole system (Figure 6). If any values fall to "2" or below, calibration is required to obtain better data accuracy. This often happens for magnetometers when magnetic/ferromagnetic or electronic devices are brought close to the BNO055 sensor.

Data read from BNO055 by the Arduino Nano microcontroller are displayed on an OLED screen and sent to MARVIN through the HC-05 Bluetooth module. Local backup is not yet implemented because of the Arduino Nano RAM limitation (Figure 6), and the device is too small for a double Arduino board solution, as for UGO.

To collect the spherical coordinates measured by UGO/Range-R, the MARVIN Android application was developed with "MIT App Inventor", an online development platform that uses a block-based programming language [54,55]. The MARVIN Graphic User Interface (GUI) was built to easily allow the addition of information about morphogenetic agents, shape type, and the cartographic coordinates of the mapper (WGS84/UTM Zone 32N).

Furthermore, it is possible to see the collected data in real time, to save the mapped shapes or to delete wrong shapes. Every record is related to a single geometry and its attributes, the format adopted is the CSV and the geometry is expressed as Well Know Text (WKT) format. The saved data are stored in three text files (.txt) located on a smartphone's internal memory, named as the WKT geometry type. Every record was saved in the proper file to separate point, linear, and polygonal geometries.

For the acquisition of measuring station coordinates, both for UGO and Range-R, a Garmin eTrex 20x single frequency GNSS receiver was used. Data was manually insert in MARVIN for each measuring station. This function will eventually be implemented directly on the devices to simplify the mapping procedure and ensure reliability, especially for Range-R which can be used dynamically without a ground-fixed tripod, as is the case for UGO.

The mechanical body structure of UGO and Range-R were an ad hoc design using "Autodesk Fusion 360" CAD (Computer-Aided Design) software and made with an "Ender 3 Pro" 3D printer.

Finally, a semi-automatic import procedure and geomorphological legend is still being implemented [56]. It counts 61 forms of 7 morphogenetic agents (Table 3) following the "guidelines for geomorphological cartography" published by ISPRA (Istituto Superiore per la PRotezione Ambientale) [57]. The symbology chosen is just a part of the whole legend cited, the selection was closely related to the geomorphological elements expected to be found during the case-study fieldwork related to the purpose of the current paper (see Section 2.4).

**Table 3.** Morphogenetic agents and shape mapped.

| Code | Morphogenetic Agent | Point | Line | Polygon | Total |
|------|---------------------|-------|------|---------|-------|
| TE | Tectonic | | 1 | 2 | 3 |
| LS | Lithostructural | | 4 | | 4 |
| GR | Gravitative | 2 | 14 | 11 | 27 |
| FD | Fluvial | | 9 | 6 | 15 |
| GL | Glacial | 1 | 9 | | 10 |
| PN | Periglacial and nival | | | 1 | 1 |
| AN | Anthropic | | 1 | | 1 |

In order to simplify the ISPRA proposed multiscale approach [57], only one geometry type for each legend element (point, line or polygon) was selected. The choice was made because the devices' maximum range does not allow for mapping all the geomorphological scales. Although this simplification has the disadvantage that the possibility of mapping geomorphological elements on very different scales is lost, it was not considered influential for the present work. Since one cannot use the GOGIRA system for mapping large-scale areas from only one measuring station, the mapper must change location to obtain a proper LOS. Furthermore, it is important to remember that the main purpose of the GOGIRA system is the mapping of real-world elements from a mapper's direct observation and provides an ability to detect small geometries which are difficult to identify on a DTM hillshade, topographic map, or satellite imagery, making GOGIRA useful for highly detailed NC.

The import procedure is very simple and uses the "Add Delimited Text Layer" QGIS import layer option, a function that allows the import of WKT shapes and relative attributes from a CSV file. Following a simple procedure (Section 2.3.4), it is possible to automatically apply the symbology categorization prepared ad hoc for the GOGIRA system. For this purpose, a library [56] was made for the symbols chosen using automatic symbol orientation and "geometry generator" such as for landslide or conoid flow direction.

### 2.3. Procedures and Data

GOGIRA's data management is summarised into four main steps: (i) field survey with UGO and Range-R devices; (ii) geometry and attribute data storage in MARVIN; (iii) cartographic coordinates calculation with CoordFinder; and (iv) QGIS data import and categorization.

#### 2.3.1. Field Survey

For correct data acquisition, the choice of the measuring station is fundamental, it is important to have a good LOS to the targets. CoordFinder is a DTM-based algorithm, so vegetation cover is not relevant (Figure 7A). However, topographic obstacles that prevent a direct LOS will compromise the data collection giving "false positives" in the CoordFinder algorithm, due to the proximity of the LOS to the topographic surface (Figure 7B). Furthermore, it is not recommended to use targets where the LOS is almost tangent to the topographical surface (Figure 7D), this can minimize the possibility of missing LOS-DTM intersection and the consequent corruption of the mapped shape. If the topographic surface surrounding the measuring station is highly irregular or with a low slope (Figure 7C), it is useful to take field notes to easily correct false intersections in the post-elaboration phase.

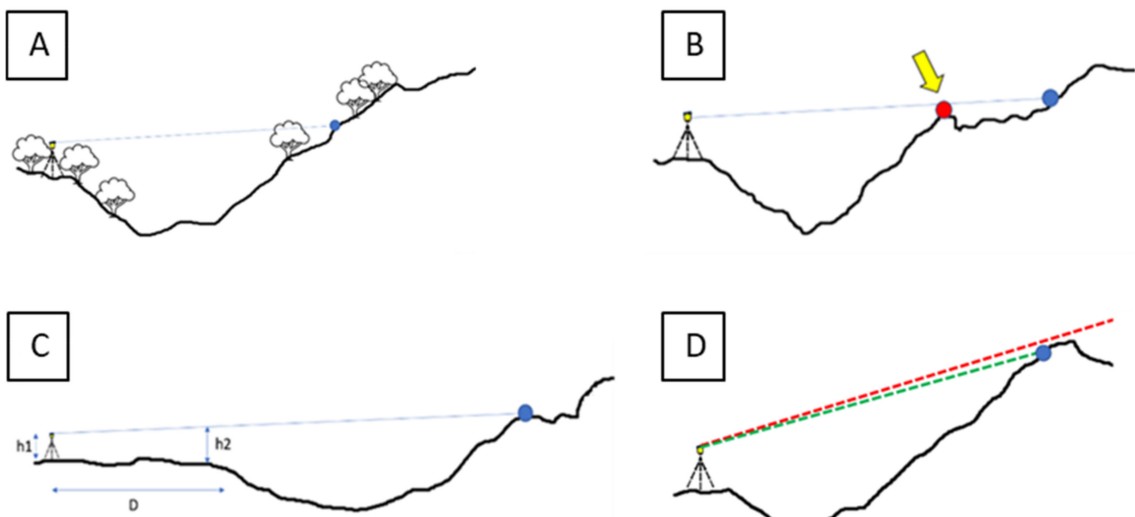

**Figure 7.** Field Stations. (**A**) Correct measurement condition with good LOS, vegetation is irrelevant. (**B**) Topological obstacle, the LOS can be too close to the topographical surface and CoordFinder can return false target coordinates. (**C**) LOS is tangent to the topographical surface, distance D and h2 are important to the CoordFinder setup buffer. (**D**) Target, too close to the relief top, can be missed with a small angle error and CoordFinder will miss the matching between LOS and topographical surface.

After the choice of a good measuring station, geomorphological mapping can be carried out using UGO or Range-R.

The first step is to use UGO as the initial setting procedure. When the device is turned on, the date is required after that the main menu is showed. An initial quick calibration is needed (Figure 8A). The main body rotational axis must be almost perpendicular to the horizontal plane, tripod asset regulations can be used, along with the toroidal bubble level, to check the horizontality of the device. After the levelling procedure, the azimuth

calibration is needed to ensure angle values with respect to magnetic north: after selecting "calibration" from the main menu, take a landmark heading with an analog compass, then with UGO, and manually set the heading value.

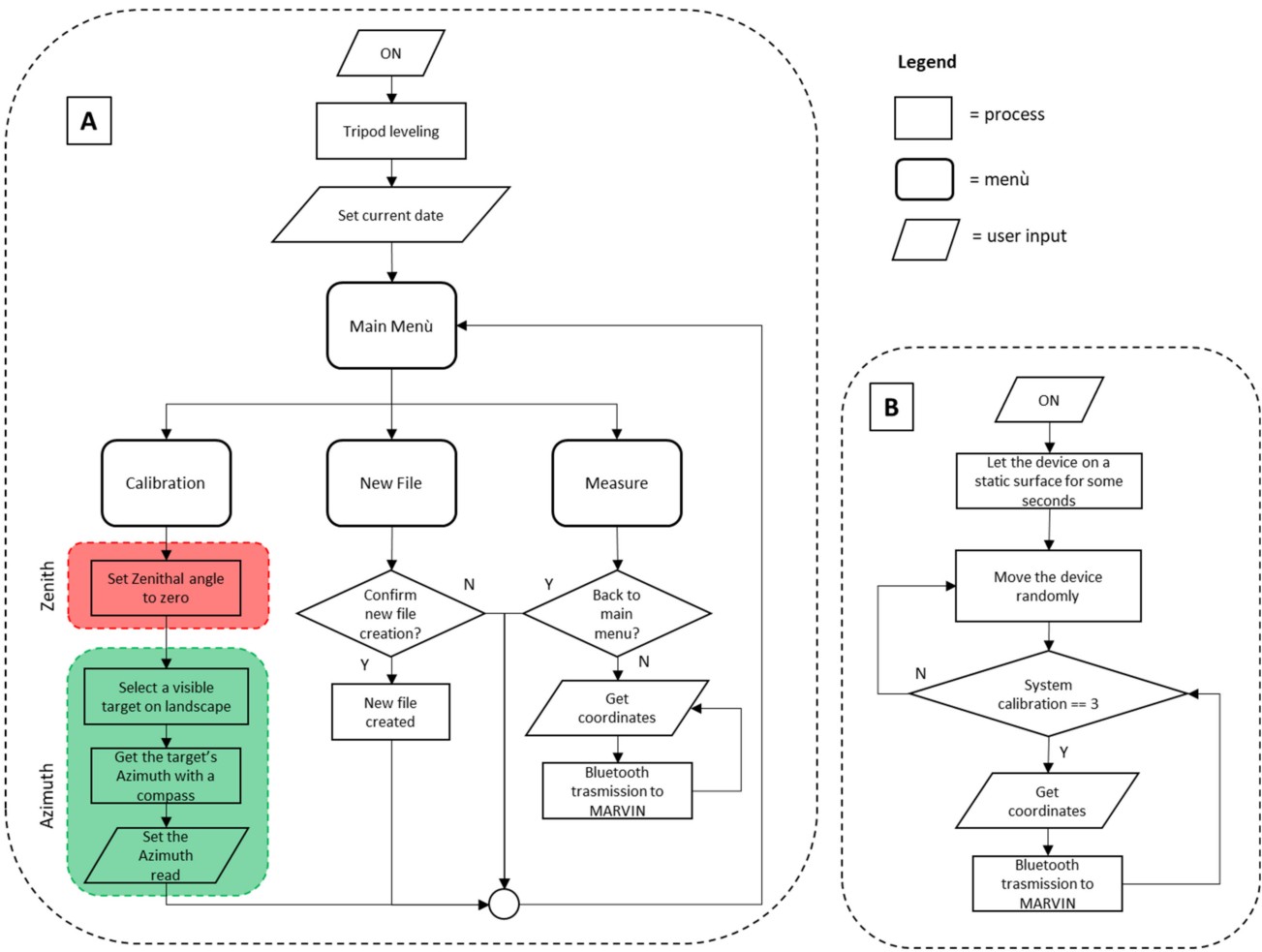

**Figure 8.** (**A**) UGO, and (**B**) Range-R workflow.

To create a backup of the working session, "new file" must be selected from the main menu. The filename is coded with a random alphanumeric header and tail based on the date set.

Finally, selecting the "measure" mode from the main menu displays the measure screen and UGO is ready for mapping. To obtain target spherical coordinates, just sight it with the optical sight and press "get" button, data will automatically be stored on the SD card and sent to MARVIN.

Range-R use is easier than UGO. Although the calibration principles are more complex, the Bosch automatic calibration algorithm makes the procedure very fast and easy (Figure 8B). When the device is switched on, it is necessary to leave the device stationary on a static surface for several seconds to calibrate the gyroscope. To calibrate the magnetometer, it is sufficient to move the device randomly to rotate the sensor around each axis. The accelerometer is automatically calibrated due to a calibration performed during the development phase of Range-R (Figure 5C). Once this procedure has been carried out, it is possible to directly acquire the data, which will be sent to MARVIN via Bluetooth.

During the measurement procedure with Range-R, it is important to keep the calibration parameters under control (Figure 6) and, if necessary, repeat the calibration procedure described.

### 2.3.2. MARVIN

Before launching the application, it is necessary to perform the Bluetooth pairing operation between the smartphone, where MARVIN was installed, and UGO or Range-R (only one device at a time). This operation must be done once for every Bluetooth device.

The MARVIN setup procedure is required at every launch of the application. The first step is the connection to UGO/Range-R device; it is important to not confuse this operation with the pairing procedure described above. Once the operation has been completed the setup panel will pop-up (Figure 9A).

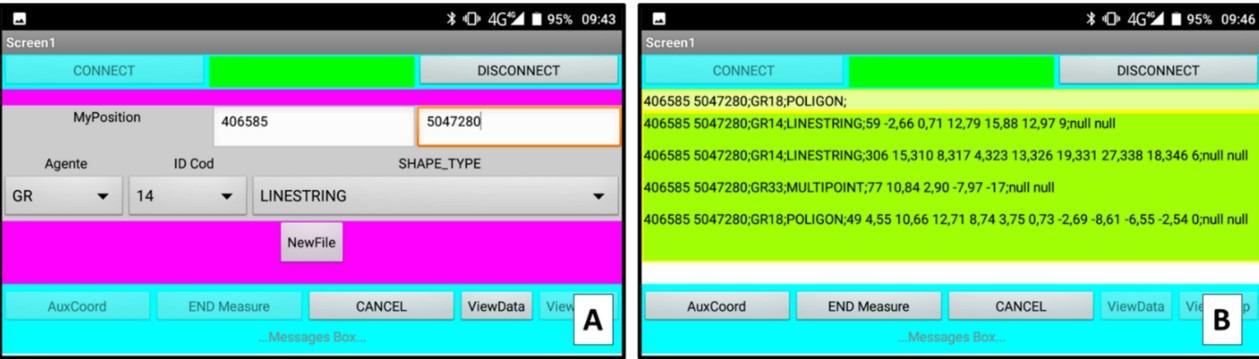

**Figure 9.** MARVIN Android App (**A**) setup screen and (**B**) measure screen.

After connection, setup data can be set, the data required are: (i) mapper position coordinates (in WGS84/UTM Zone 32N), (ii) morphogenetic agent code (Table 3), geomorphological element code [57], and (iii) shape geometry type (Table 4). At the current state of development, the GNSS receiver is not yet installed on the devices, so east and north coordinates of the measuring station must be inserted by user.

**Table 4.** WKT format.

| Geometry | WKT Format |
| --- | --- |
| Point | MULTIPOINT(P1x P1y, P2x P2y, . . . , Pnx Pny) |
| Linear | LINESTRING(P1x P1y, P2x P2y, . . . , Pnx Pny) |
| Polygonal | POLYGON((P1x P1y, P2x P2y, . . . , Pnx Pny, P1x P1y)) |

When setup is completed, the "NewFile" button can be pressed. This initiates the measurement session of the specific geomorphological element set in the setup and data acquisition with UGO/Range-R can start. Data obtained by the devices are shown in real time and an audio signal confirms the correct data acquisition by MARVIN.

At the end of the measurement session, an additional point may be necessary for the semi-automatic symbology categorization. In this case, press the "AuxCoord" button and acquire the needed point, otherwise by pressing the "EndMeasure" button the current shape measure will end, and data will be saved. To map a second shape, albeit with the same setup parameters of the precedent mapped, a "New File" session must be started with the procedure described above.

To monitor the data acquisition process, it is possible to switch between the "data" panel and the "setup" panel using the dedicated buttons (Figure 9A,B). This operation will not interrupt the data collecting procedure if a shape was being mapped. If any problem arises during a measurement session, it is possible to press the "CANCEL" button to delete the current measurement; this does not involve the loss of data previously saved.

### 2.3.3. CoordFinder

The algorithm was designed to run in a standalone executable file (.exe) with a Graphic User Interface (GUI) using a Python library (Table 2). Inputs required (Figure 10) are DTM,

data input, and data output file paths. The DTM extent (WGS84/UTM Zone 32N) is expressed as minimum and maximum values of east and north coordinates; the resolution is in meters; and magnetic declination of the measuring station is measured in degrees from magnetic north. An additional "buffer" parameter is required, it is the range zone of no-data intersection from the measuring station coordinates. This is necessary to avoid the topographic adverse condition (see Section 2.3.1). The CoordFinder working field notes about the surrounding landscape are strongly recommended for a properly set buffer parameter.

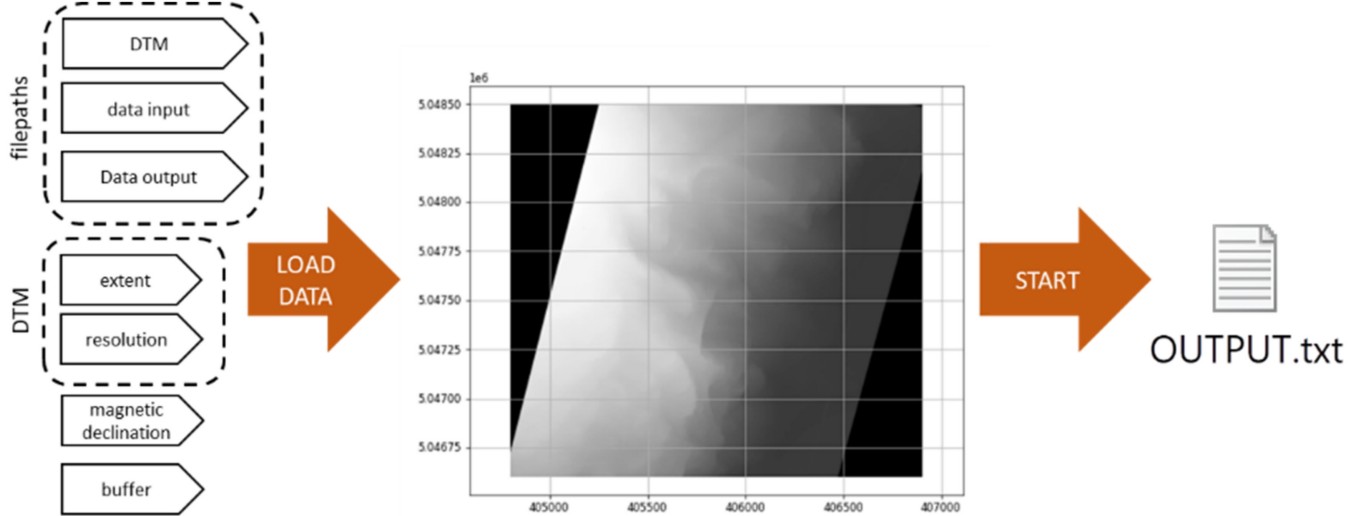

**Figure 10.** CoordFinder data requirements and operative steps.

By pressing the "LOAD DATA" button, the data setup will be uploaded and the DTM shown on the GUI (Figure 10). To launch CoordFinder just click the "START" button and the algorithm will run for each point of every geomorphologic element present in the loaded data file. Execution time can strongly vary; it is related to the DTM size, the number of targets sighted, and PC performance.

Output data will be saved in WKT format in the "output file directory" (Figure 10), the saving process is made one-by-one for each shape. This means, that if the algorithm fails (e.g., bug or inconsistent input data), the shape successfully processed will not be lost.

### 2.3.4. QGIS Data Import

After checking that the project coordinate system is UTM/WGS-84, the file generated by CoordFinder algorithm can be imported into QGIS with the "Add Delimited Text Layer" command. This procedure is very fast and advanced computer skills are not required. Due to the WKT geometry format and CSV file format, the shapefile can be imported, and symbology categorized.

As for data import, the procedure (Figure 11A) is as follows: (i) check that the reference coordinates are set to UTM/WGS-84; (ii) click on "Add Delimited Text Layer", then (iii) select the file generated by CoordFinder; (iv) from the "File Format" menu, select the "Custom delimiter" item and enable the "Tab" checkbox; (v) in the "Geometry Definition" menu, select the "Well known text (WTK)" box, select "Geometry" in the "Geometry field" and "Detect" in "Geometry type", then (vi) click the "Add" button, the geometries will be loaded with their attributes.

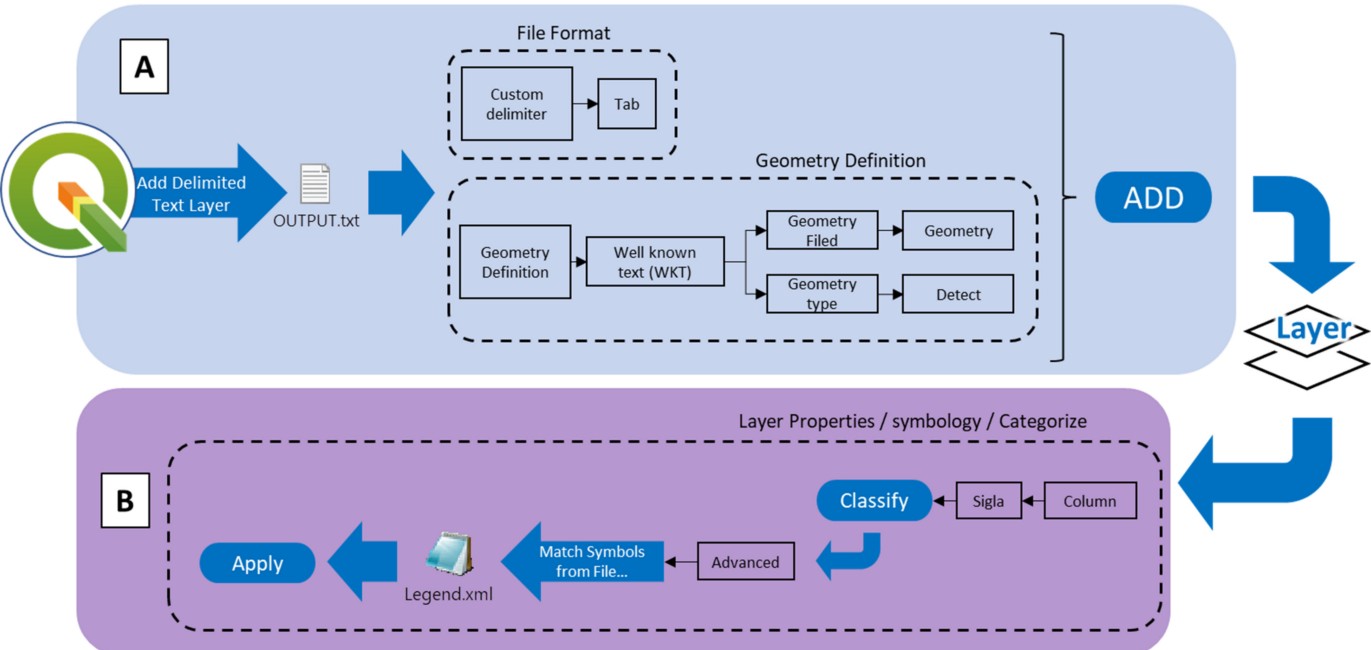

**Figure 11.** QGIS (**A**) "Add Delimited Text" import procedure, and (**B**) semi-automatic categorization procedure.

The semi-automatic categorization procedure (Figure 11B) requires the geomorphological legend library specially made for GOGIRA project. The file can be freely downloaded from GitHub [56]. After that, (i) opening the layer properties panel in the (ii) Symbology sub-panel (iii) the Categorized option can be chosen, and the file (iv) classified by "Sigla" attribute. Then (v) select the "Advanced" menu and "Match to Symbols from File", select the file path where the Geomorphological Legend was pasted, and (vi) press the "Apply" button to apply the symbology. This procedure can be applied to every kind of shapefile, the only requirement is that the "Sigla" attribute field is in accordance with the Geomorphological Legend and the "AuxCoord", for some geomorphological elements.

*2.4. Field Test: Quincinetto Landslide System*

To evaluate GOGIRA, a complex landslide system in northwestern Italy (Figure 12A) was chosen as a test site. It is in Piemonte region near the Val d'Aosta region boundary, a few kilometres north of the town of Quincinetto. The area is sparsely populated but well-studied and monitored [19] because of its close proximity to an important highway (A5) that connects the Piedmont and Aosta Valley regions; as well as being a crossing point for France and Switzerland.

The field test area was chosen considering (i) the high quality of available data (Figure 12B–D), (ii) the difficult accessibility related to the high slope (elevation from about 290 to 1250 m a.s.l.) (Figure 12B,C), and landslide risk. The first point is fundamental for a good qualitative comparison to validate the map results, while the second point is a perfect application for a remote mapping system, such as GOGIRA.

Considering the early development state of the GOGIRA system, a semi-qualitative evaluation approach was chosen. It was based on the comparison between GOGIRA's final outputs and the highly detailed data from landslide system monitoring. The main purpose of the field test was the practical evaluation of the CoordFinder algorithm, due to its central role in the GOGIRA system. UGO was chosen as the reference data acquisition tool due to the reliability of its simple-operating-principles sensors, while with Range-R only few data were collected for a preliminary qualitative evaluation.

Three different tests were conducted to:

- Estimate the mapping precision with varying distance from the measuring station and targets (metric difference);
- Check the morphometric coherence between the mapped shapes and the land morphometry (graphical comparison);
- Evaluate GOGIRA's final mapping result by comparison with a highly detailed geomorphological map made with modern tested methods (maps comparison).

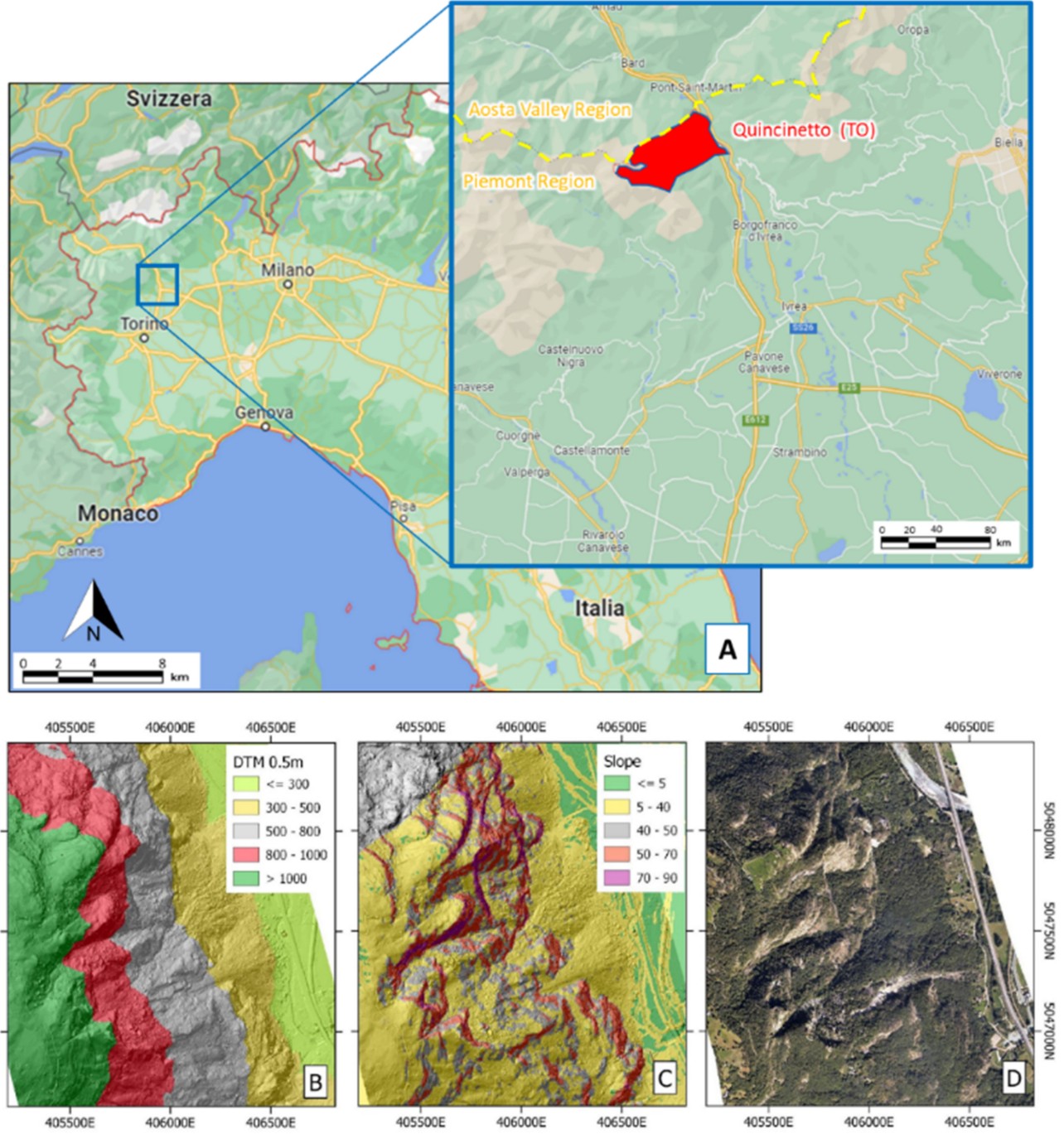

**Figure 12.** Quincinetto settings. (**A**) General geographic position of the landslide system. High-resolution (**B**) DTM, (**C**) Slope, and (**D**) orthophoto of the landslide system.

## 3. Application and Results

### 3.1. Data Acquisition and Elaboration

For the Quincinetto area, the measuring stations are chosen to test GOGIRA under different conditions. One (SM_01) from a far distance (up to 2700 m) to obtain a general view of the investigated slope, two (SM_02 and SM_03) at the bottom of the landslide system to map highly-detailed features, and one (SM_04) on the upper sector of the landslide system (Figure 13). Distances from included targets are between a minimum of 280 metres and a maximum of 2700 m (Table 5).

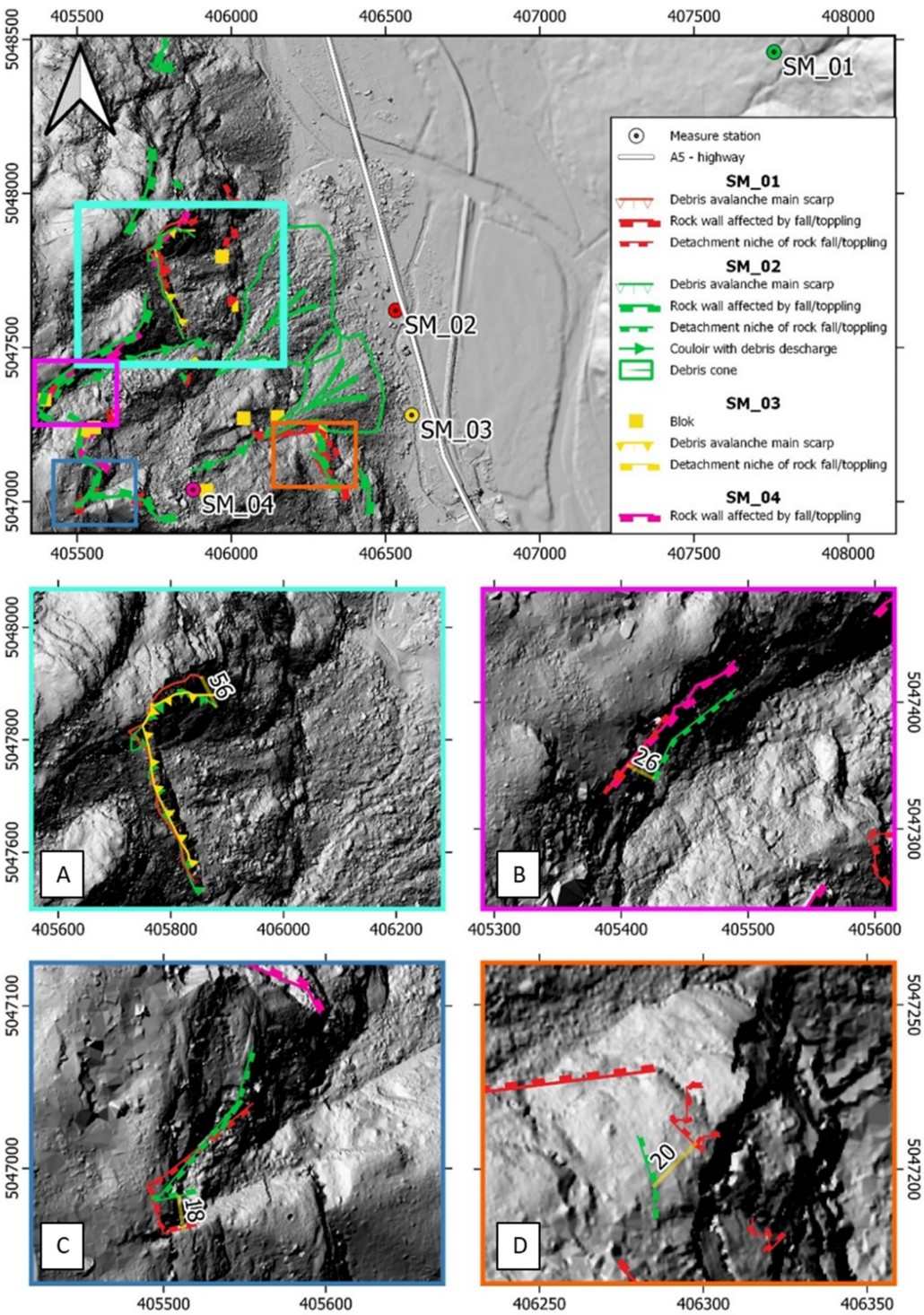

**Figure 13.** (**A–D**) Metric difference between shape mapped from stations.

**Table 5.** Measure stations and shapes.

| Station | East [1] | North [1] | Min Distance [m] | Max Distance [m] | Device | Geometry | N° |
|---------|----------|-----------|------------------|------------------|--------|----------|-----|
| SM_01 | 407760 | 5048459 | 1560 | 2700 | UGO | Line<br>Polygon | 25<br>2 |
| SM_02 | 406533 | 5047620 | 450 | 1200 | UGO | Line | 24 |
| SM_03 | 406585 | 5047280 | 670 | 1000 | UGO<br>Range-R | Point<br>Line | 9<br>4 |
| SM_04 | 405888 | 5047016 | 280 | 890 | UGO | Line | 4 |

[1] Reference system: WGS84/UTM Zone 32N.

The elements mapped in the test sessions include 68 morphometries, subdivided as shown in Table 5. With Range-R, only a few elements were mapped in the first mapping test phase because its greater complexity can introduce an additional uncertainty. The main objective was the CoordFinder algorithm test.

Data collection conducted on site SM_01, SM_02, and SM_03 required about four hours, including time for moving between stations. The SM_04 was difficult to access and logistically required more time, although data collection is the fastest step in the procedure.

To test CoordFinder algorithm, two DTMs were used. The first one was a 5-meterre-solution DTM from a LiDAR survey of the entire Piemonte region ("Ripresa aerea ICE 2009–2011") [58]; the second was a 0.5 m resolution DTM made ad hoc with an aerial LiDAR survey specially-made [19] to study and monitor the landslide system. The 5-m DTM was finally chosen for the test because results with both DTMs were comparable, and the ICE 2009–2011 is an official data available for a wider region. The 0.5-m resolution DTM was only used for high resolution hillshade morphometric comparisons and graphical map presentation. An important consideration was to test GOGIRA under standardized conditions and not limited to a specific case study.

Angular coordinates were successfully processed with the standalone executable. CoordFinder converted collected angular coordinates to the UTM/WGS84 reference system and saved as WKT files. These outputs files were imported into QGIS and categorized with described procedures (Section 2.3.4).

*3.2. Metric Difference*

Factors that influence the GOGIRA system precision are numerous, such as: UGO/Range-R's angular accuracy and precision, GNSS error, DTM resolution, azimuth calibration and/or drift, human error, field survey visibility (illumination, vegetation), and finally CoordFinder algorithm computations. Considering this, a comparative approach was adopted to evaluate the precision's distance dependency. This was chosen to understand the possible range of application for a survey with GORIRA. Results were not expected to be a definitive precision evaluation, but a good practical estimation of the prototypes aided by geomorphologist experience.

Data collected by UGO/Range-R were processed with CoordFinder and imported into a QGIS project. Geomorphological elements were checked to find shapes mapped both from a long distance (SM_01) and medium/short distance (SM_02, SM_03, and SM_04) (Figure 13). Metric distance between shapes were measured using the QGIS "measure tool". Some significant examples are shown in Figure 13B–D. Results highlight a variable misfit, from a minimum of 18 m to a maximum of 56 m. The best fit between GOGIRA's mapped shapes and the geomorphological map was found in the macro-scale elements (rock avalanche niche, scarps affected by rock fall/toppling).

As explained in Section 2.4, shapes mapped with Range-R are not numerous enough to provide a proper estimation. Furthermore, it is interesting to notice how the rock avalanche niche (Figure 13A) mapped with Range-R from SM_03 fit in between that mapped with UGO from SM_01 and SM_02.

### 3.3. Graphical Comparison

To develop a coherent map in a QGIS project, all the data collected with the GOGIRA system were integrated, merging the geomorphological elements mapped from every measurement station. Elements mapped by multiple stations were cleaned to eliminate redundant shapes (Figure 14), the choice was made based on geomorphological experience, to obtain the best results. The final map count includes a total of 42 geometries: 9 points, 31 polylines, and 2 polygons, subdivided as shown in Table 6.

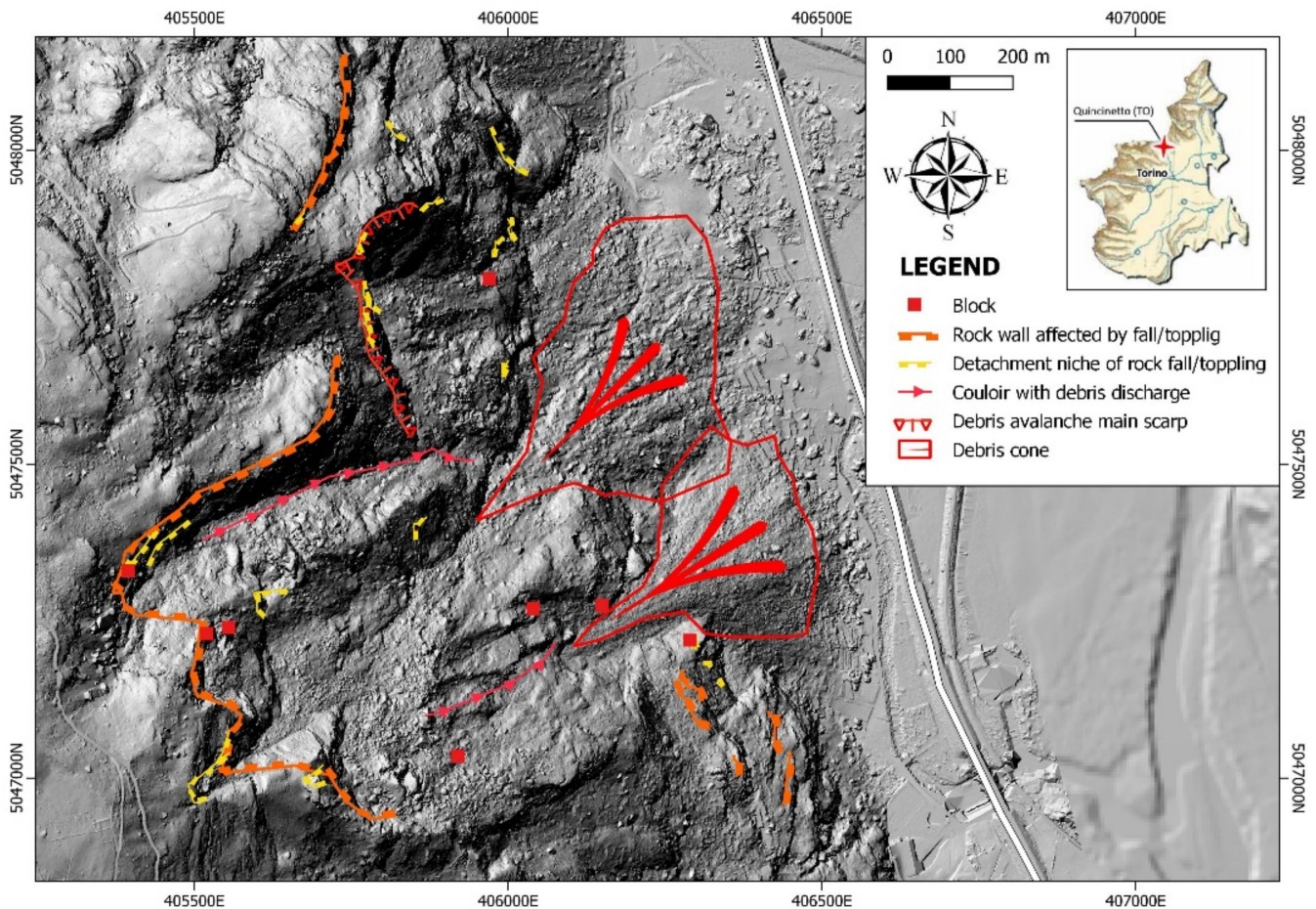

**Figure 14.** GOGIRA's geomorphological map.

**Table 6.** Total mapped elements after cleaning process. The "Agent" and "Element" fields are referred to the MARVIN setup (see Section 2.3.2).

| Element | Agent | Element | Geometry | N° |
|---|---|---|---|---|
| Block | GR | 33 | Point | 9 |
| Detachment niche of rock fall/toppling | GR | 15 | Line | 21 |
| Rock wall affected by fall/toppling | GR | 14 | Line | 7 |
| Debris avalanche main scarp | GR | 08 | Line | 1 |
| Couloir with debris discharge | GR | 18 | Line | 2 |
| Debris cone | GR | 34 | Polygon | 2 |

The legend symbology colours were slightly different with respect to the ISPRA directive for the Automatic Legend because the high density of shapes created confusion between "scarps affected by rock fall/toppling" and singles "niche of fall/toppling". Often when symbols were overlayed it resulted in graphical confusion.

The landscape morphology was represented by hillshades made with QGIS tools from available DTMs (0.5- and 5-m resolution), contours from the same DTMs, and a 1:10 k scale CTR (Carta Tecnica Regionale) topographic map [58].

Morphometries mapped with GOGIRA, and cleaned, were overlayed on the morphometric datafile available as shown in Figure 15. This was made to (i) check the morphometric validity of the GOGIRA system, and (ii) to understand if the geomorphological elements sighted directly in the field were visible and mappable with commonly used topographical data (hillshade and CTR).

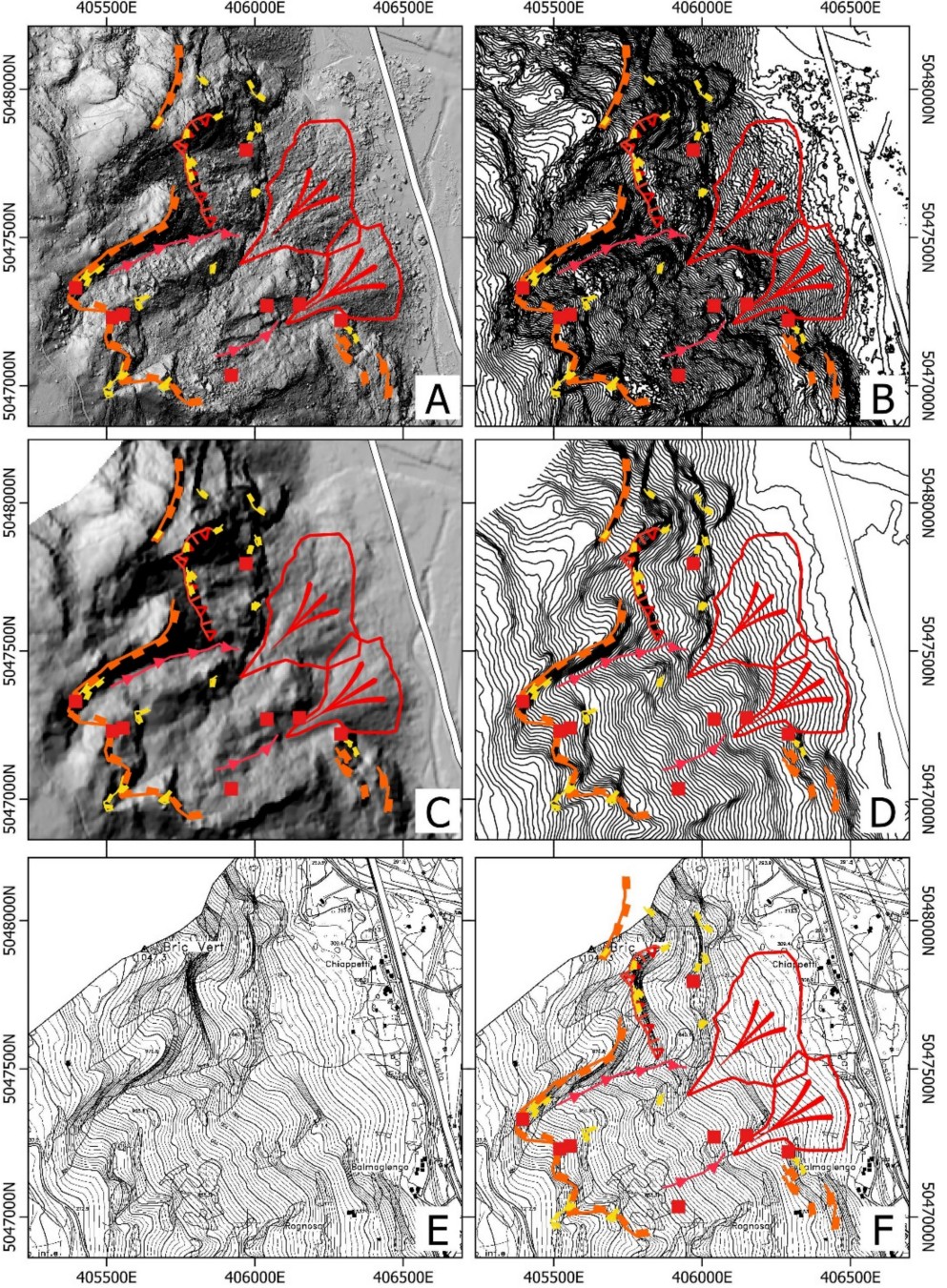

**Figure 15.** Graphic comparison between GOGIRA's geomorphological map and available topographical data. (**A**) Hillshade from 0.5 m DTM. (**B**) 10 m contour from 0.5 m DTM. (**C**) hillshade from 5 m DTM. (**D**) 10 m contour from 5 m DTM. (**E**) CTR map 1:10 k scale only. (**F**) CTR map 1:10k scale with GOGIRA's geomorphological map.

As shown in Figure 15A, a variability for the fitting between GOGIRA mapped elements and topographical surface, is evident. The linear elements observable at the macroscale, generally have a good fitting (GR14, GR08, GR18) probably due to the extension and the major relief on the 5-m DTM used for CoordFinder. With regard to the meso-scale linear elements (GR15), there is more uncertainty. This is due to the relatively small dimension of morphometries, often not recognizable on 5-m DTM hillshades, and not even on the CTR. From the hillshade, contour map, and CTR, it is not possible to identify the single block (GR33) element mapped with GOGIRA. For the debris cones (GR34), both the 5-m and 0.5-m hillshades provide precision for identification of the morphometry, while the shapes mapped with GOGIRA are not the best fitting (Figure 15A). This is because the survey was conducted in the spring season when the vegetation is still dense and direct LOS with the ground is often not possible. The results were an incorrect overlay of the conoids.

The morphological/geomorphological interpretation, using DTMs, contours and CTR, showed some important limitations. With respect to the GOGIRA system that can digitalize an observed geomorphological element, topological data have only information about the land morphometry. Furthermore, CTR and DTM (and its derivatives) are simplifications of the reality, as is true of all digitalized data. This is less relevant for high-resolution data, such as 0.5-m hillshade, but have a significant role for the 5-m hillshade and 1:10k CTR.

It is evident that the CTR (Figure 15E,F) is not sufficiently detailed to properly identify the morphometries mapped with GOGIRA. This means that using a CTR-topographical map for a geomorphological survey is not possible for mapping real-world elements visible in the field.

### 3.4. Maps Comparison

The final test was a direct cartographic comparison between the GOGIRA map and a highly-detailed geomorphological map made by previous authors [19] with well-established techniques such as: (i) photogrammetric and LiDAR survey, (ii) geomorphological field survey, (iii) comparative analysis of geomorphological and geological-technical data, and (iv) morpho-evolutionary model of the slope. Results are shown in Figure 16.

Geomorphological and GOGIRA maps were overlayed with QGIS to highlight the differences. As shown in Figure 16, there is an excellent correspondence of the linear elements. Furthermore, with GOGIRA, it is possible to map a significant number of geomorphological elements. This was possible because, for each mapped shape, the morphogenetic agent that created it, is required. Direct attribution bypasses the problem of categorizing morphometry by hillshade and contour. Although morphometry is evident from these data, it is often difficult to link it to the correct morphogenetic agent. On the other hand, the GOGIRA system maps the geomorphological elements visible in the field and directly categorizes them.

As for the polygonal elements, only two debris cones were mapped (with UGO, from SM_01). It is evident that an underestimation of the area of one cone and the actual geometrical overlap wasn't identified. The debris avalanche accumulations were difficult to recognize from SM_01, due to the dense vegetation, and inability to map from the other stations; the detrital talus was not visible from the stations chosen and was thus not mappable.

Regarding point elements, only single blocks were mapped with GOGIRA. They were not mapped with traditional methods because of the impossibility of direct access (for GNSS mapping) to the high-risk area.

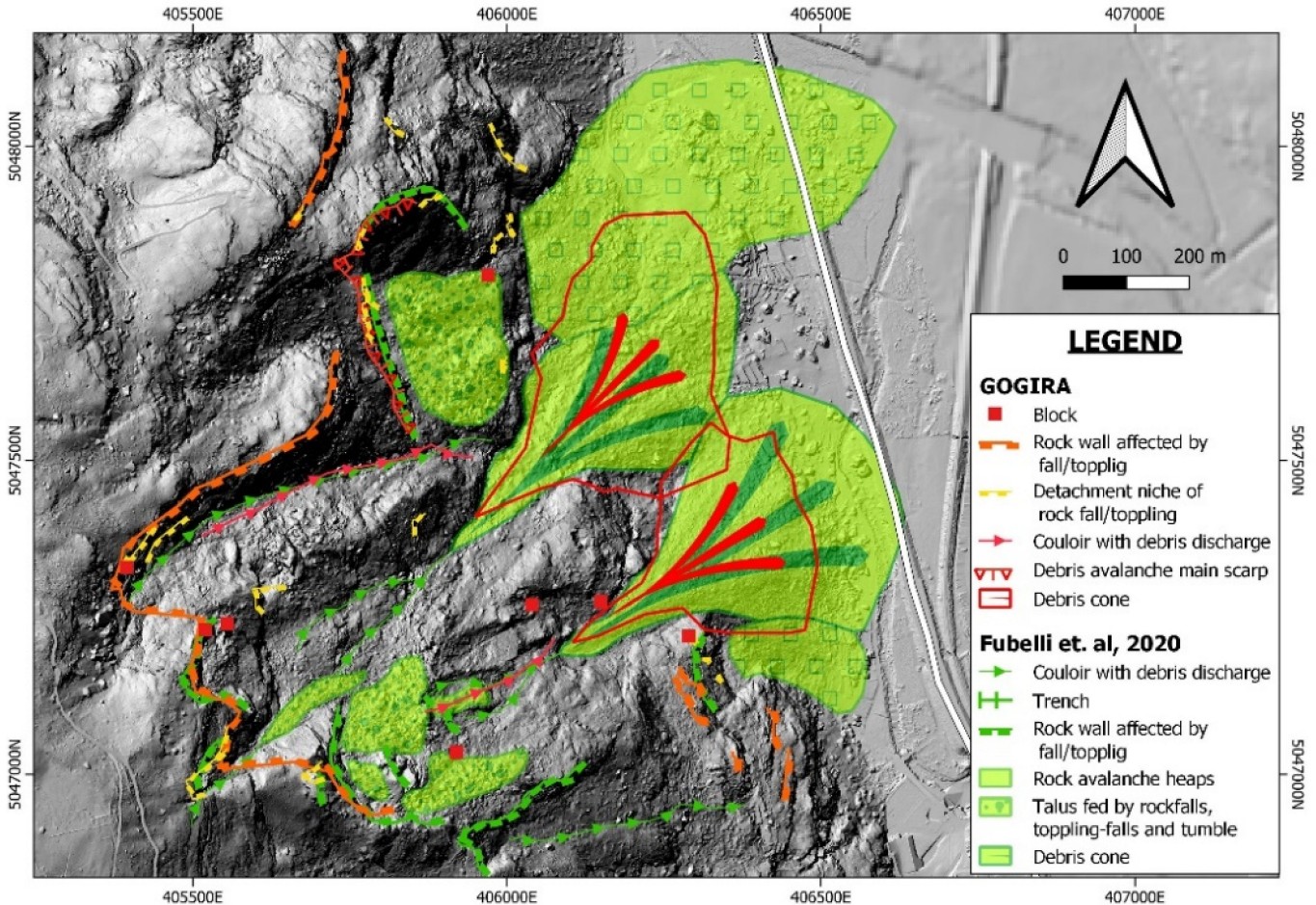

**Figure 16.** Map comparison.

## 4. Discussion

The Quincinetto landslide system, chosen as a field test area, had the perfect conditions to perform a preliminary qualitative evaluation of both GOGIRA reliability and application in risky conditions. Steep slope and high rockfall risk are key problems for geomorphological mapping survey, both for accessibility and safety. Furthermore, the exposition of the A5 highway to landslide hazard make the case study a practical application for the torrential planning of the GOGIRA system.

Measuring stations were chosen to test the system at different distances: from a minimum of 280 m (SM_04) to a maximum of 2700 m (SM_01). The SM_04 highlights the good response of the CoordFinder algorithm in processing data, also in poor measurament conditions (low visibility, difficult access, irregular topography around the measuring station). On the other hand, a point-of-view so close to the slope was a serious obstacle to the correct geomorphological interpretation of the landscape due to a loss of the overall view. As for the SM_01, the far distance did not support mapping the smallest geomorphological elements (mapped from SM_02 and SM_03) but was perfect for the general mapping of the whole landslide system, even from the opposite slope of the valley, with respect to the investigated area.

With the technological evolution, high-resolution DTMs have become much more accessible [10], however the spatial cover of these data is discontinuous. For this reason, a 5-m resolution DTM was chosen as input to CoordFinder instead of the 0.5-m resolution. The first was developed for the entire Piemonte territory from a LiDAR survey, while the second was an ad hoc development for the small area of Quincinetto landslide system. This difference became crucial when the GOGIRA system was designed to make a more accessible and easy DNC technique.

As introduced in Section 2.4, tests conducted were focused on a semi-qualitative practical approach to evaluate the reliability of GOGIRA in a practical mapping survey. Tests targets were: (i) the precision—distance relation estimation (metric differences), (ii) the check of the morphometric coherence (graphical comparison), and (iii) the comparison of the final mapping results with modern tested mapping methods (map comparison).

Results of the metric comparison showed that the elements mapped from SM_01 versus SM_02 and SM_03 have a varying misfit, from a minimum of 18 m to a maximum of 56 m. As shown in Figure 13, the best fitting were the geomorphological elements at the macro scale, such as rock avalanche niche, and scarps affected by rock fall/toppling.

From the graphical and map comparison there was a good fitting of the macro-scale linear elements with the DTM morphometry. While, for some meso-scale elements, their small morphometric size was difficult to accurately represent by CoordFinder because the 5-m DTM was not precise enough. The debris cones mapped with GOGIRA demonstrated an incorrect overlay of the shapes (Figure 14). This happened because of the dense vegetation, which doesn't allow for a direct LOS with the ground.

With GOGIRA it was possible to map a major number of elements with respect to INC methods used by previous authors [7,19]. This was possible due to the possibility of direct mapping of geomorphological elements visible in the field and directly categorizing them. On the other hand, polygonal shapes were only partially mapped and with less accuracy as compared to the geomorphological map made by Fubelli et al. [19].

## 5. Conclusions

The evolution of Geomorphological Numerical Cartography techniques had a significant but slow improvement during the last decades [7,9]. This slowness is largely due to the lack of practical-use and economic devices and methodologies [20,23,24].

The GOGIRA system was created to make a contribution to the Direct Numerical Cartography methodologies and techniques. The definition of GIS proposed by Ershad et al. [4] was adopted to develop a user-friendly system that integrate hardware, software, procedures, and data.

The aim of the GOGIRA system was the realization of an accessible DNC system that allows for safely generating reliable geomorphological maps in landslide risky conditions, with low-cost components and user-friendly procedures.

The system is still in the prototype phase, but currently consists of: (i) two devices (UGO and Range-R) for field remote data acquisition, (ii) a Control Unit (MARVIN) for data collection and categorization, (iii) an algorithm (CoordFinder) for data elaboration, and (iv) semi-automatic procedures for QGIS import and categorization.

Finally, what emerged from this study are the following highlights:

- DNC can improve and optimize geomorphological mapping;
- A GIS-structured project can be used to developed new methods for DNC with standardized and interconnected devices and software;
- GOGIRA proved to be a valid system for geomorphological DNC applied to a complex landslide system. Considering the early stage of development, results were excellent for mapping linear and point objects, as for polygonal elements, more studies must be conducted to improve accuracy and precision;
- Finally, for both INC and DNC, high-resolution DTMs are fundamental for a good quality, detailed geomorphological map, while CTR is often not suitable for mapping meso or micro-scale elements.

In this research GOGIRA was tested for landslide mapping, however other fields of application can be explored. The key points that must considered for further developments are the necessity of good Line Of Sight (Figure 7), a valid DTM of the area, and a MARVIN software update, since it was designed specifically for geomorphological mapping.

**Author Contributions:** Conceptualization, M.L. and G.F.; methodology, M.L. and G.F.; software M.L.; validation, M.L. and G.F.; formal analysis, M.L. and G.F.; investigation, M.L. and G.F.; resources M.L. and G.F.; data curation, M.L. and G.F.; writing—original draft preparation, M.L. and G.F.; writing—review and editing, M.L. and G.F.; visualization, M.L. and G.F.; supervision G.F.. All authors have read and agreed to the published version of the manuscript.

**Funding:** This research received no external funding.

**Data Availability Statement:** Data presented in this study are available on request from the corresponding author. These are not publicly available because the dataset is part of unpublished research data and therefore not publishable.

**Acknowledgments:** We want to thank Laurie Jayne Kurilla for the English review. A special thanks to Stefano Faga for the developing assistance.

**Conflicts of Interest:** The authors declare no conflict of interest.

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
