# Peer review of "The GOGIRA System: An Innovative Method for Landslides Digital Mapping"

_geosciences, doi:10.3390/geosciences12090336_

Round 1
Reviewer 1 Report
In this paper, a new system following the GIS (Geographic Information System) scheme is proposed. It is a suite of hardware and software tools, algorithms, and procedures to make easier and cheaper DNC. Initial tests conducted on the Quincinetto landslide system (north-western Italy) demonstrated good results in terms of morphometric coherence and precision. The reviewer thinks the topic discussed in this paper is very important, which is of great significance for landslide mapping. This reviewer sees that a minor revision will be needed before being accepted for possible publication. Here are the main comments for the revision.
(1) A key problem in this paper is the lack of introduction to the applicability of the proposed method. Is this model applicable to all regions?
(2) The introduction section lacks some important references, such as:
Centrifugal model test on a riverine landslide in the Three Gorges Reservoir induced by rainfall and water level fluctuation; Prediction of landslide displacement with step-like behavior based on multialgorithm optimization and a support vector regression model. A novel seepage device and ring-shear test on slip zone soils of landslide in the Three Gorges Reservoir area.
(3) Some Figs are unclear and unexplained. For example, figure 7 is not clear. Figure 9A lacks key information such as scale and compass.
(4) The conclusion is not concise and innovative. I believe that the Authors should try to interpret and explain more clearly their results. Some key quantitative conclusions should be supplemented.
(5) In this study, the authors must improve the statements about what is new in their study and what are the contributions to the developments of agriculture.
(6) The text is not clear. The author is suggested to check the professional vocabulary, grammar and spelling of the full text.
Reviewer 2 Report
The paper shows a new system for an innovative method for landslides digital mapping. The introduction describes several systems for aerial mapping of urban aspect, showing potential application. To improve this part in aerial infrared thermography at urban level is suggested the recent review https://doi.org/10.1007/978-981-19-1894-0_1. It could enlarge the possibility of this technique for digital mapping at urban scale. The sims of the paper are not introduced. It is necessary to introduce them in section 2, also showing the novelty of the system and the differences with other similar systems. The description of the system is clear. In the contrary, results are not clear. They seem more application of the systems rather then results. Introduce a section with the description of the field analysis, aims, area, features, and so on. Maps. Imparino s x interesting session, but it could be more interesting using different methods described in the introduction.
Reviewer 3 Report
Excellent research. The manuscript has some linguistic issues which can be improved by review through professional services. I leave that up to the editor and the authors. I believe addition of few flowcharts to express the overall workflow in addition to the figure 7 will add some value to the manuscript. However, it is up to the authors if they agree with me or not. The manuscript can be accepted in present for with minor changes.
Round 2
Reviewer 2 Report
-